# Going Beyond State-Reaching:
# Learning Abstractions for Intrinsically Motivated Option Discovery

## Abstract

Temporal abstraction via options can improve exploration in vast environments. However, existing option discovery algorithms find subgoals that target all aspects of the state simultaneously. This *state-reaching* approach produces options that only apply in narrow regions of the state-space, eventually causing an explosion in the number of options that overwhelms the agent, and impedes progress on its primary task of reward maximization. We introduce an algorithm that instead identifies a small, relevant subset of features for each subgoal, yielding options that generalize broadly and accelerate exploration. Our approach learns abstract, transferrable options and achieves rapid exploration in three sparse-reward, image-based domains, including the Atari game MONTEZU­MASREVENGE.

## 1. Introduction

Reinforcement learning (RL) agents must interact with the environment at every timestep. Yet, effective decision-making may demand that the agent build temporal abstractions to ease central challenges of credit assignment, exploration, and transfer (Klissarov et al., 2025). The core question is—how can RL agents autonomously discover temporally extended actions, or *options* (Sutton et al., 1999), solely via environment interaction? This problem, known as skill discovery or option discovery (Sutton et al., 1999), is the core bottleneck for scaling hierarchical RL (HRL; Barto and Mahadevan, 2003) methods to solve long-horizon, high-dimensional problems (Konidaris, 2019; Gershman, 2017; Sutton et al., 2022). However, the vast majority of skill discovery algorithms today have a common shortcoming—they build options that resort to the state-reaching objective; we illustrate this through the following example.

[1]Anonymous Institution, Anonymous City, Anonymous Region, Anonymous Country. Correspondence to: Anonymous Author <anon.email@domain.com>.

Preliminary work. Under review by the International Conference on Machine Learning (ICML). Do not distribute.

Recall successfully riding your bicycle for the first time. The moment you balanced your bicycle without training wheels, you realized that you had done something right. While practicing this newly acquired skill in the future, you recreated your speed and balance, not the color of your t-shirt, the weather conditions outside or the exact wear and tear on your tires. However, when existing skill discovery algorithms attempt to learn a policy to recreate a previously encountered experience (Chentanez et al., 2005), they attempt to recreate *every aspect* of that state. Examples include the $\epsilon$-ball from deep skill graphs (Bagaria et al., 2021b), pixel-wise equality (Veeriah et al., 2018), the perceptual hashing from Go-Explore (Ecoffet et al., 2021), random projections of the target state (Dabney et al., 2021; Farebrother et al., 2023), or a neural classifier that predicts perceptual similarity between states (Vezhnevets et al., 2017; Hafner et al., 2022). This state-reaching strategy adversely impacts hierarchical agents in several ways. First, the resulting skills are nontransferable; consider a robot grabbing a cup from a cluttered table: the skill's subgoal is tied to the position of every object in the state, requiring the agent to learn combinatorially more skills. Beyond stymieing transfer, which is a key benefit of HRL (Taylor and Stone, 2009), state-reaching artificially reduces the size of the skill's subgoal region, which complicates policy learning and eventually makes the skill less useful. Finally, recreating a state is *unscalable* because in vast, realistic domains, recreating all aspects of a particular state may simply be impossible (Bagaria and Schaul, 2023; Javed and Sutton, 2024).

Our skill discovery algorithm is inspired by intrinsic motivation (Oudeyer et al., 2008; Colas et al., 2022): during play, when children cause something interesting to happen, they try recreate it, with increasing efficiency (White, 1959). They practice this skill until they get bored and move on, but retain the ability to reuse the acquired skill later (Barto et al., 2004). Similarly, when during an AI agent's exploration, it notices something particularly interesting, it should learn a skill to reliably achieve it in the future. Other algorithms (for example, Bagaria et al. 2021b; 2025; Jinnai et al. 2020) use these ideas, but resort to a state-reaching objective. Instead, we *identify the subset of the state features responsible for the novelty* (Chentanez et al., 2005) of that state, and create a subgoal achievement classifier that attends only to

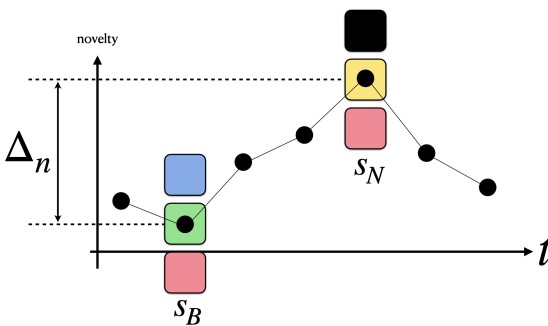

Figure 1: **Illustration of the core insight.** Suppose an agent observes a state trajectory of length 7 and a novelty estimator assigns each state $s_t$ in that trajectory a novelty score $f_\phi(s_t)$. $s_N$ denotes the most novel state, and $s_B$ denotes the most boring state in that trajectory. Suppose that each state has 3 features; the colored boxes represent feature values for those two states. Rather than simply treating $s_N$ as a target state, we seek to identify which features in particular were responsible for the large difference in novelty $\Delta_n = f_\phi(s_N) - f_\phi(s_B)$. Our algorithm can handle input images, but we show state features here for simplicity.

those specific features[1]; this core insight is illustrated in Figure 1. These classifiers serve as pseudo-reward functions for training option policies, which can be composed by a high-level policy over options to maximize extrinsic reward.

We test our algorithm in three challenging exploration problems: MINIGRID-KEYCORRIDOR (Chevalier-Boisvert et al., 2023), VISUALTAXI (Dieterich, 2000; Allen et al., 2021), and the Atari game MONTEZUMASREVENGE (Bellemare et al., 2013)—all with image-based observations, and sparse reward functions. We find that our agent finds truly abstract subgoals—i.e., subgoals that only depend on a small number of features and are achieved in qualitatively different states.

## 2. Background and Related Work

We model the agent-environment interaction as a Markov Decision Process $M = (S, A, R, T, \gamma)$ (Puterman, 1994; Sutton and Barto, 2018). To aid in reward maximization, our agent will discover options $o \in \mathcal{O} = (\mathcal{I}_o, \pi_o, \beta_o)$ (Sutton et al., 1999), where $\mathcal{I}_o \subseteq S$ is the initiation set, $\pi_o$ is the intra-option policy over primitive actions, and $\beta_o : S \to \{0, 1\}$ is the termination condition. Each option has a maximum horizon of $H$ steps.

**Subgoal Options.** While options need not optimize any objective in general, for option *discovery* it is convenient to

---
[1]We do not assume that state features are given and learn subgoal achievement classifiers directly from images, but the language of "features" helps present our ideas simply.

think of options as achieving subgoals—indeed, most option discovery algorithms can be viewed this way (Precup, 2001; Bagaria, 2025; Klissarov et al., 2025). A *subgoal option* uses the termination condition $\beta_o$ to serve three roles: it defines the subgoal (the set of states where $\beta_o(s) = 1$), it determines when the option terminates, and it provides a sparse pseudo-reward $R_o(s) = \beta_o(s)$ for training $\pi_o$. The initiation set $\mathcal{I}_o$ consists of states from which the option has a high probability of achieving its subgoal. This formulation reduces option discovery to identifying useful subgoals: once $\beta_o$ is specified, standard policy optimization can train $\pi_o$, and policy evaluation can train $\mathcal{I}_o$ (Bagaria et al., 2023).

**Universal Value Function Approximators (UVFAs)** (Schaul et al., 2015) are a scalable way to learn goal-conditioned value functions $V_g : S \to \mathbb{R}$, $Q_g : S \times A \to \mathbb{R}$ and their corresponding goal-conditioned policies $\pi_g : S \to A$ (Kaelbling, 1993) using function approximation. We use UVFAs to parameterize option value functions and policies: instead of representing each option's policy separately (Sutton et al., 2011), all options can condition the same function approximator on its own subgoals (Bagaria et al., 2021a): $\pi_o(s) = \arg\max_a Q_g(s, a) = \arg\max_a Q_{\beta_o}(s, a)$.

**Novelty-based exploration.** In tabular RL, count-based exploration bonuses that decay as $1/\sqrt{N[s, a]}$ can guide efficient learning (Strehl and Littman, 2008), but do not scale to large state spaces. Pseudocounts (Bellemare et al., 2016) generalize counts by assigning similar values to similar states; Coin Flip Networks (CFN; Lobel et al., 2023) are a simple, state-of-the-art technique for estimating pseudo-counts from images. While novelty-driven agents demonstrate impressive performance (Kapturowski et al., 2022), they do not learn options, which are key for transfer (Taylor and Stone, 2009) and high-level planning (Konidaris et al., 2018; Sutton et al., 2024). We seek to bridge this gap by using novelty as the substrate for option discovery.

**Feature attribution via Shapley values.** When multiple features jointly produce an outcome, how should we attribute credit to each one? Shapley values (Shapley, 1953) provide a principled answer from cooperative game theory. The key idea is to measure each feature's *marginal contribution*—how much the outcome changes when that feature is added—averaged over all possible orderings of features. Concretely, for a function $f$ and a set of features $\{1, \ldots, k\}$, the Shapley value of feature $i$ is:

$$\phi_i = \frac{1}{k!} \sum_\sigma \left[ f\left(S_i^\sigma \cup \{i\}\right) - f\left(S_i^\sigma\right) \right], \qquad (1)$$

where the sum is over all permutations $\sigma$ of features and $S_i^\sigma$ is the set of features appearing before $i$ in permutation $\sigma$. Intuitively, this asks: if features were added one at a time in a random order, how much would feature $i$ contribute on

average? Computing exact Shapley values requires evaluating $f$ on exponentially many subsets, which is intractable for high-dimensional inputs such as images. So, we use the DeepShap algorithm (Lundberg and Lee, 2017; Shrikumar et al., 2017), which efficiently approximates Shapley values for neural networks. DeepShap computes attributions by measuring the network output changes for the input versus some reference baseline, then assigns credit to each input feature by propagating this difference back through the network layers using local linear approximations of the activation functions. This provides pixel-level attributions that approximate Shapley values for each pixel's contribution to the network output.

**Option discovery methods.** Option discovery techniques can be broadly categorized by their learning signal. Reward-driven methods like Option-Critic (Bacon et al., 2017) and feudal approaches (Dayan and Hinton, 1993; Vezhnevets et al., 2017) learn options using extrinsic reward, but struggle in sparse-reward settings. Empowerment-based methods (Eysenbach et al., 2019; Gregor et al., 2016) learn diverse skills that increase the agent's control over its environment, showing promise in exploration (Hansen et al., 2021; Campos Camúñez et al., 2020). Spectral methods use the graph Laplacian to find principal directions of the state-space for exploration (Machado et al., 2017; Jinnai et al., 2019; Klissarov and Machado, 2023). Some other algorithms also combine goal-conditioned RL and novelty (Pitis et al., 2020; Pong et al., 2019; Simsek and Barto, 2004; Bagaria et al., 2025). While all these approaches differ in their objectives, they share a common limitation: skills resort to state-reaching because they attend to all features. Our work is orthogonal—we focus on *how* subgoals are represented, showing that identifying relevant features enables broader generalization. For a detailed survey of option discovery, we refer readers to the work of Klissarov et al. (2025).

**Learning Subgoal Achievement Classifiers.** Most option discovery algorithms either require manually defined subgoal functions (Ecoffet et al., 2021) or resort to state-reaching (Veeriah et al., 2018), where the benefits of abstraction fade (Colas et al., 2022). Methods that learn classifiers typically still attend to all state features: perceptual similarity approaches threshold distances in observation space (Andrychowicz et al., 2017), use random projections (Dabney et al., 2021) or autoencoders (Tang et al., 2017); temporal distance methods predict whether states are within $K$ steps of a goal (Savinov et al., 2018; Mendonca et al., 2021); and mutual information methods like DISCERN (Warde-Farley et al., 2019) abstract away uncontrollable features but not irrelevant, controllable ones. None explicitly identify a small set of relevant features for each subgoal.

**Options for feature attainment.** The strategy of learning options that target specific features has been explored in HRL (Hengst, 2002). STOMP (Sutton et al., 2024) learns options that each maximize a single state feature, though it is unclear why maximization is the right objective or why options should be limited to one feature. Proto-goal RL (Bagaria and Schaul, 2023) addresses this by learning options that achieve target values for combinations of features. However, both methods assume a factored state representation is provided, while our approach discovers the relevant feature set autonomously from interaction data.

## 3. Method: Abstract Subgoal Discovery

Modern RL systems can efficiently compute proxies for intrinsic motivation, like novelty (Kapturowski et al., 2022). However, their use as a guiding signal for option discovery focuses on state-reaching, which fails to scale beyond modest navigational problems (Hansen et al., 2021). Instead of using novel states directly as option subgoals (Simsek and Barto, 2004; Bagaria et al., 2025), our agent discovers options by isolating which features of a novel state are responsible for its novelty. When the agent encounters a particularly novel state $s_N$ during exploration, it compares $s_N$ to the least novel state $s_B$ in the same trajectory and asks: which features of $s_N$ account for the novelty difference $\Delta_n = f_\phi(s_N) - f_\phi(s_B)$? Features whose modification substantially changes $\Delta_n$ are retained; the rest are discarded (Figure 2). A classifier is then constructed to attend only to these relevant features, serving as an abstract subgoal. The agent learns an option policy to achieve this subgoal, and the cycle repeats: the agent explores from promising subgoals, discovers new novel states, and creates new options.

Our agent repeatedly executes the following high-level steps: (a) identify the subgoal with the highest potential for exploration, (b) achieve that subgoal by executing the corresponding option, (c) explore from that subgoal after achieving it, (d) identify the most (locally) interesting state in that trajectory, (e) determine which features make that state interesting, (f) initialize a new option to achieve the new subgoal, and add it to the agent's set of options.

### 3.1. Option Selection and Exploration

Our agent maintains a set of options $\mathcal{O}$, each defined by a subgoal classifier $\beta_o$ discovered during exploration. At each episode, the agent selects a subgoal to explore from, navigates there using the corresponding option policy, and then executes a novelty-seeking policy.

**Policy over options.** Not all subgoals are equally useful for exploration: some are unreachable from the current state, others lead to already-explored regions. The policy over options $\pi_\mathcal{O}(o \mid s)$ balances achievability with exploration

potential. First, we restrict selection to options the agent can reliably achieve by defining the initiation condition:

$$\mathcal{I}_o(s) = \mathbb{I}\{V_{\beta_o}(s) > \delta\}, \tag{2}$$

where $\delta$ is a threshold hyperparameter. Among options for which $\mathcal{I}_o(s) = 1$, we score each by the expected value of exploring from its subgoal:

$$U(o) = \mathbb{E}_{s' \sim \beta_o}\big[V_{\text{CFN}}(s')\big], \tag{3}$$

$$\text{where } V_{\text{CFN}}(s') = \mathbb{E}_{\pi_{\text{CFN}}}\left[\sum_t \gamma^t\big(r_t + \lambda f_\phi(s_t)\big)\right], \tag{4}$$

and $\lambda$ controls the tradeoff between extrinsic reward and intrinsic novelty. The policy samples proportionally to utility:

$$\pi_{\mathcal{O}}(o \mid s) = \frac{U(o)}{\sum_{o':\mathcal{I}_{o'}(s)=1} U(o')}. \tag{5}$$

**Conducting exploration.** Once an option $o$ is sampled from $\pi_{\mathcal{O}}$, the agent executes its intra-option policy $\pi_o$ to reach the subgoal. Upon success, the agent executes a CFN policy $\pi_{\text{CFN}}$ (Lobel et al., 2023) that maximizes cumulative novelty and reward, producing a trajectory $\tau = (s_1, \ldots, s_K)$.

**Detecting novel events.** If any state in $\tau$ is sufficiently novel, the agent initiates subgoal creation. Specifically, if $\max_{s \in \tau} f_\phi(s) > \mu + \sigma$, where $\mu$ and $\sigma$ are the running mean and standard deviation of novelty scores, the agent extracts $s_N = \arg\max_{s \in \tau} f_\phi(s)$ and $s_B = \arg\min_{s \in \tau} f_\phi(s)$, then passes them to the feature selection algorithm (Section 3.2).

### 3.2. Identifying Relevant Features

Given $s_N$ and $s_B$ from the previous section, we identify which features of $s_N$ explain the novelty difference $\Delta_n = f_\phi(s_N) - f_\phi(s_B)$. We decompose this into two steps: extracting candidate features from the image observation, and then measuring each feature's contribution to $\Delta_n$.

**Feature extraction.** Neural representations learned end-to-end are entangled, making it difficult to intervene on individual factors (Rodriguez-Sanchez et al., 2025; Kumar et al., 2025). We instead use computer vision techniques to extract discrete, interpretable features. In visually simple domains (MINIGRID, VISUALTAXI), we subtract a background image and detect contours (Suzuki and Abe, 1985). In visually complex domains (MONTEZUMASREVENGE), we use a pre-trained Segment Anything Model (SAM; Kirillov et al., 2023) to extract object masks. Both approaches yield bounding boxes $\{b_1, \ldots, b_k\}$ representing candidate features.

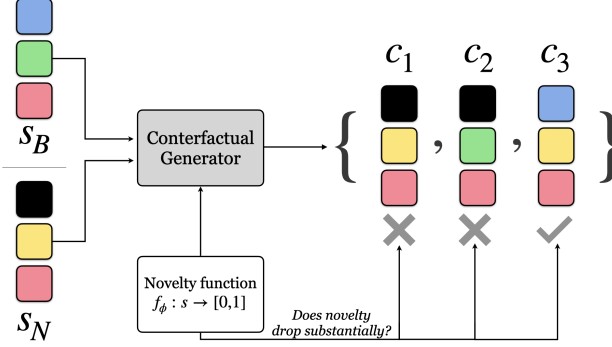

Figure 2: **Illustration of our feature selection algorithm.** The most boring and most novel states, $s_B$ and $s_N$ respectively, are input into the Counterfactual Generator, which outputs counterfactual states $c_1, c_2, c_3$—each with one feature reset to its value in $s_B$. Only resetting the first feature (black → blue) substantially lowers novelty, indicating that the first feature accounts for most of the change in novelty $\Delta_n$. This one-at-a-time reset is effective when features contribute independently. In domains with complex feature interactions, we instead use DeepSHAP to compute each feature's marginal contribution to $\Delta_n$, accounting for interactions between features.

**Feature attribution.** Given candidate features, we must determine which ones explain the novelty difference. The core question is: how much does each feature $b_i$ contribute to $\Delta_n$? We consider two approaches, both grounded in the same intuition—a feature is relevant if changing it substantially affects the novelty prediction.

*Counterfactual reset.* For each feature $b_i$, we construct a counterfactual image $c_i$ by replacing the patch at $b_i$ in $s_N$ with the corresponding patch from $s_B$. If $\Delta_n = f_\phi(s_N) - f_\phi(c_i) > \epsilon$, then $b_i$ is relevant. This simple approach is effective when features contribute to $\Delta_n$ roughly independently.

*DeepSHAP attribution.* When features interact, we use DeepSHAP (Lundberg and Lee, 2017; Shrikumar et al., 2017) to approximate Shapley values. DeepSHAP efficiently computes these attributions by backpropagating through the novelty network $f_\phi$, decomposing the difference $\Delta_n = f_\phi(s_N) - f_\phi(s_B)$ into per-pixel contributions $\phi_p$. We aggregate within each bounding box: $\text{Shapley}(b_i) = \sum_{p \in b_i} \phi_p$. Bounding boxes exceeding threshold $\mu_\phi + \sigma_\phi$ (mean and standard deviation of pixel attributions) are retained.

Both methods yield a subset of features $\mathcal{F} \subseteq \{b_1, \ldots, b_k\}$ that explain the jump in novelty from $s_B$ to $s_N$. These features define the scope of the subgoal classifier constructed in the next section.

### 3.3. Subgoal Classifier Construction

Most of $s_N$ is irrelevant to why it was novel. The feature selection step identified which parts matter; everything else can vary freely and be ignored. The classifier we construct respects this—it checks only whether the relevant features $\mathcal{F}$ match their values in $s_N$, and ignores the rest.

We construct a non-parametric classifier $g : S \rightarrow \{0, 1\}$ since $s_N$ is our only positive example. The classifier stores $s_N$ and a feature extractor $h$ that returns only the relevant features—selecting relevant indices for factored states, or cropping bounding boxes for images. Given a state $s$:

$$g(s) = \mathbb{I}\Big\{ \mathcal{D}\big(h(s_N), h(s)\big) < \xi \Big\}, \qquad (6)$$

where $\xi$ is a threshold and $\mathcal{D}$ measures dissimilarity (e.g., Euclidean distance for factored states, cross-correlation for images).

### 3.4. Option Learning

The classifier $g$ serves as the new option's termination condition $\beta_o$ and provides a sparse terminating pseudo-reward $R_o(s) = g(s)$ for training. The intra-option policy $\pi_o$ is trained online using goal-conditioned R2D2 (Kapturowski et al., 2019) with hindsight experience replay (Andrychowicz et al., 2017). The initiation function $\mathcal{I}_o(s)$ is derived from the goal-conditioned value function $V_g$, as described in Section 3.1. The new option is added to $\mathcal{O}$ and becomes available for future use by the policy over options $\pi_{\mathcal{O}}$.

### 3.5. Putting It Together

Algorithm 1 summarizes the full option discovery loop. The agent alternates between exploiting existing options to reach promising subgoals and exploring to discover new ones. When exploration yields a sufficiently novel state, the agent identifies the relevant features, constructs a new subgoal classifier, and adds the corresponding option to its repertoire.

## 4. Experiments

Our experiments address three questions: (1) Does our feature attribution method correctly identify the features responsible for novelty? (2) Do the resulting subgoal classifiers generalize across states that differ in irrelevant features? (3) Does this abstraction lead to better exploration and task performance compared to flat RL and state-reaching HRL methods?

### 4.1. Experimental Setup

We evaluate on three sparse-reward, image-based domains where the agent must explore effectively without relying on dense reward signals.

---

**Algorithm 1** Abstract Subgoal Option Discovery

1: **Initialize:**
2: Initialize CFN to learn novelty function $f_\phi$, value function $V_{\text{CFN}}$, and policy $\pi_{\text{CFN}}$.
3: Initialize goal-conditioned R2D2 to learn value function $Q_g$ and policy $\pi_g$.
4: Initialize option set $\mathcal{O}$ with a default exploration option.
5: **while** True **do**
6:     Sample option $o \sim \pi_{\mathcal{O}}(o \mid s)$ using Equation 5.
7:     Execute $\pi_o$ until $\beta_o(s) = 1$ or timeout.
8:     **if** timeout **then**
9:         **continue** {Resample option}
10:     **end if**
11:     **if** agent reached subgoal **then**
12:         Roll out $\pi_{\text{CFN}}$ to get trajectory $\tau$.
13:     **end if**
14:     **if** $\max_{s \in \tau} f_\phi(s) > \mu + \sigma$ **then**
15:         Extract $s_N = \arg\max_{s \in \tau} f_\phi(s)$.
16:         Extract $s_B = \arg\min_{s \in \tau} f_\phi(s)$.
17:         Extract candidate features $\{b_1, \ldots, b_k\}$ from $s_N$ (Section 3.2).
18:         Select relevant features $\mathcal{F}$ using counterfactual reset or DeepSHAP.
19:         Construct classifier $g$ from $\mathcal{F}$ (Section 3.3).
20:         Create option $o' = (\mathcal{I}_{o'}, \pi_{o'}, \beta_{o'} = g)$ and add to $\mathcal{O}$.
21:     **end if**
22:     Update $Q_g$ and $\pi_g$ using R2D2 and HER.
23:     Update $\pi_{\text{CFN}}$ to maximize reward $(r_t + \lambda f_\phi(s_t))$.
24: **end while**

---

**Domains.** MINIGRID-KEYCORRIDOR-S5R3 (Chevalier-Boisvert et al., 2023) requires the agent to navigate through rooms, pick up a key, and unlock a door. Colas et al. (2022) identified it as a challenging exploration problem due to its sparse rewards and combinatorial state space. VISUALTAXI (Dieterich, 2000) requires navigating to a passenger, picking them up, and delivering them to a destination. Variants of this problem have been widely used to evaluate HRL algorithms (Dieterich, 2000; Allen et al., 2021). In both domains, the agent receives a sparse reward only upon completing the entire task—no intermediate rewards for picking up the key or passenger. MONTEZUMASREVENGE (Bellemare et al., 2013) is a canonical exploration benchmark with a natural hierarchical structure: skills like climbing ladders, jumping between platforms, and collecting items can be reused across different rooms and we seek to build option discovery algorithms that can autonomously find this hierarchical structure from experience.

**Method variants.** As mentioned in Section 3.2, we use the counterfactual reset method for feature attribution in MINIGRID and VISUALTAXI. In MONTEZUMASREVENGE,

where visual complexity makes patch-based counterfactuals less reliable, we use SAM for feature extraction and DeepSHAP for attribution.

**Training procedure.** In MINIGRID and VISUALTAXI, subgoal discovery and option learning occur jointly online—exactly as described in Algorithm 1. In MONTEZUMASRE-VENGE, we use a two-stage approach to stabilize training: first, we run novelty-driven exploration to discover subgoals; then, we train an HRL agent that learns intra-option policies and the policy over options using the discovered subgoals. The learning curves in Figure 6 show only the second stage, making this a rough comparison rather than a claim about state-of-the-art performance.

### 4.2. Qualitative Results

We first examine whether our method correctly identifies relevant features and whether the resulting classifiers generalize across visually distinct states.

**Feature attribution.** Figure 4 shows two examples of DeepSHAP attribution in MONTEZUMASREVENGE. In (a), the agent enters a new room for the first time; the attribution correctly highlights the agent's position in the novel area, ignoring static elements like platforms and ladders. In (b), the agent picks up a key; the attribution focuses on the key's location. In both cases, the method identifies semantically meaningful features—entering a room, obtaining an item—rather than attending to the entire state.

**Classifier transfer.** Figures 3 and 5 demonstrates that classifiers built from relevant features transfer across contexts. In Figure 5, the leftmost image shows the state in which a classifier was learned; the bounding box indicates that it attends only to the player's position in a specific screen region. The remaining images show states where this classifier fires: despite differences in room layout, score, number of lives, and enemy positions, the classifier correctly triggers whenever the player appears in the target region. This is the genuine abstraction we seek—the same subgoal is achievable in many visually distinct states.

**Failure cases.** Not all discovered classifiers are useful. In some cases, the attribution identifies features that are novel but not reachable (e.g., a moving enemy). In others, the threshold for feature selection is too permissive, resulting in classifiers that attend to multiple features and rarely fire. We discuss these limitations further in Section 5.

### 4.3. Quantitative Results

We compare our method (Abstract Subgoals) against two baselines across all domains: R2D2 (Kapturowski et al., 2019), a flat RL method with $\epsilon$-greedy exploration; and

CFN (Lobel et al., 2023), a flat RL method with novelty-based exploration. In MONTEZUMASREVENGE, we include an additional baseline, Pixel Equality, which uses the same discovered subgoals as our method but constructs state-reaching classifiers that attend to the entire image. Figure 6 shows learning curves across all three domains.

**MiniGrid and VisualTaxi.** In both domains, Abstract Subgoals consistently solves the task while CFN and R2D2 fail to make meaningful progress. Although CFN's count-based exploration provides intrinsic reward throughout learning, it does not discover reusable structure. Abstract Subgoals, by contrast, constructs options that decompose the task into achievable subproblems.

**MontezumasRevenge.** Abstract Subgoals outperforms CFN, R2D2, and, crucially, Pixel Equality. Since Abstract Subgoals and Pixel Equality use the same discovered subgoals, this comparison isolates the effect of our classifier construction: attending to relevant features rather than entire states enables options to transfer across rooms. Due to our two-stage training procedure (Section 4.1), this is a rough comparison and we do not claim state-of-the-art performance; the result demonstrates the value of abstraction over state-reaching.

**Classifier generalization.** We measure how many unique states each classifier triggers on in MONTEZUMASRE-VENGE. On average, classifiers fire in 353.8 unique states (SE = 79.3, SD = 1292.9). The high variance reflects that some classifiers are highly selective (e.g., picking up a specific key while having exactly one life) while others are broadly applicable (e.g., the player appearing in a screen region). This confirms that our classifiers generalize beyond the single state from which they were learned.

## 5. Discussion and Conclusion

We presented an algorithm for discovering abstract subgoal options. Rather than treating novel states as targets to be recreated in their entirety, our agent identifies which features of a novel state are responsible for its novelty and constructs classifiers that attend only to those features. The resulting subgoals are abstract: they can be achieved in states that look different but share the same relevant feature values. Our experiments show that this abstraction enables more efficient exploration and better task performance than both flat RL methods and a state-reaching HRL baseline.

**Limitations and future work.** Our method relies on external tools for feature extraction—contour detection in simple domains, SAM in complex ones. While these tools are general-purpose, they impose a bias toward object-centric features. Learning disentangled state representations end-

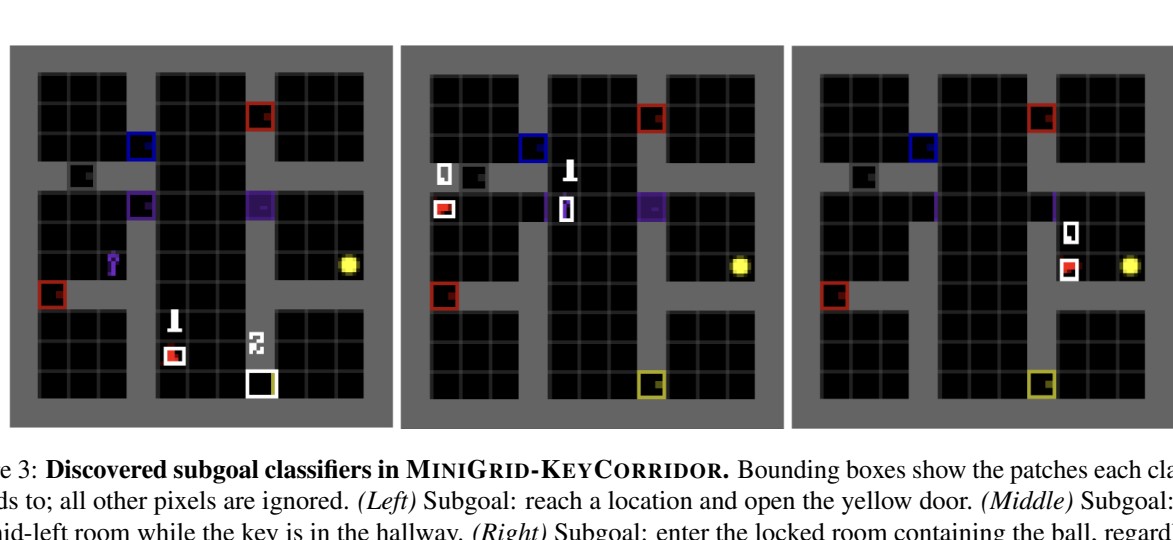

Figure 3: **Discovered subgoal classifiers in MINIGRID-KEYCORRIDOR.** Bounding boxes show the patches each classifier attends to; all other pixels are ignored. *(Left)* Subgoal: reach a location and open the yellow door. *(Middle)* Subgoal: reach the mid-left room while the key is in the hallway. *(Right)* Subgoal: enter the locked room containing the ball, regardless of key position or door configuration.

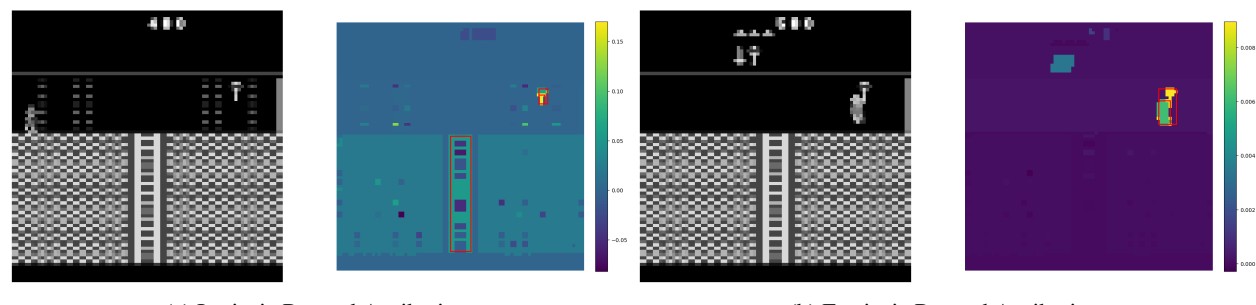

(a) Intrinsic Reward Attribution.            (b) Extrinsic Reward Attribution.

Figure 4: **Examples of identifying relevant features in MONTEZUMASREVENGE.** For each state (black and white image), we show the Shapley value attributed to each pixel in the image and the bounding boxes selected for the subgoal classifier in the image to their right. The first classifier checks if the agent has entered room 7, while the second classifier checks if the key in that room has been picked up.

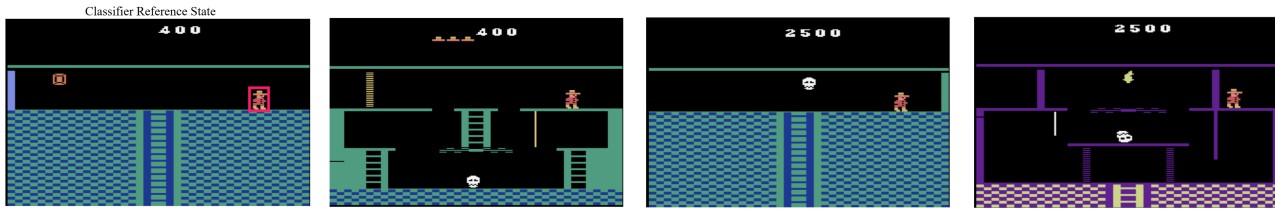

Figure 5: **Example of a discovered subgoal classifier that transfers to many different rooms.** The leftmost image shows the context in which this subgoal classifier was learned—the bounding box shows that the classifier only attends to the player's position. The remaining three images show example states in which that classifier was triggered—as long as the player re-appears in that portion of the screen, other factors such as the room, score, number of lives, location of the skull, etc, are all irrelevant for achieving that option's subgoal.

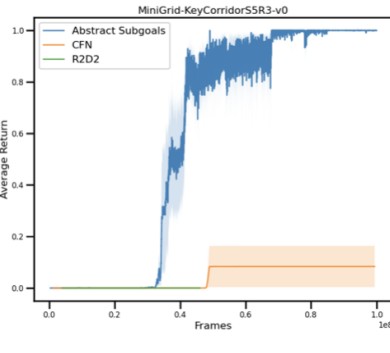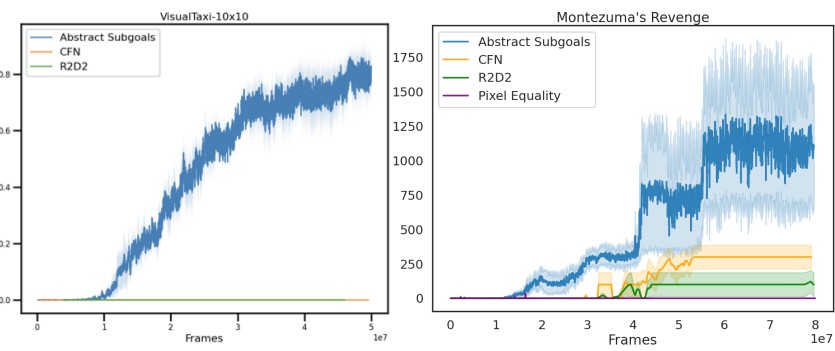

Figure 6: **Learning curves** comparing our agent (Abstract Subgoals) with a non-hierarchical novelty maximizing RL method (CFN) and a vanilla RL method (R2D2). In MONTEZUMASREVENGE, we include a state-reaching HRL baseline (Pixel Equality). Solid lines denote average undiscounted task return; shaded regions denote standard deviation (5 random seeds in the first two, 4 random seeds in the third).

to-end (Rodriguez-Sanchez et al., 2025; Kumar et al., 2025) could obviate the need for such tools (Silver et al., 2021). Not all discovered subgoals are useful: some attend to uncontrollable features like moving enemies. Off-policy methods for estimating goal controllability (Bagaria and Schaul, 2023) could filter these out before classifier construction. Our policy over options is currently a bandit that selects based on immediate utility; replacing it with a full RL policy, or abstract model-based-planning (Sutton et al., 2024; Bagaria et al., 2025), capable of multi-step lookahead could enable more strategic option sequencing. Finally, the two-stage training procedure in MONTEZUMASREVENGE limits our claims about sample efficiency; integrating subgoal discovery with online learning in visually complex domains remains open.

The key insight of this work is simple—when something interesting happens, figure out *why* it was interesting, not just *that* it was interesting. By grounding option discovery in feature attribution, we take a step toward agents that build genuinely abstract, reusable skills from experience.

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
