# OpenReview forum: "Going Beyond State-Reaching: Learning Abstractions for Intrinsically Motivated Skill Discovery"
_ICML.cc/2026/Conference — Submitted to ICML 2026_

### Official Review · Reviewer_XqSX · 2026-03-11

**Soundness:** 3
**Presentation:** 1
**Significance:** 2
**Originality:** 3
**Overall Recommendation:** 3
**Confidence:** 3

**Summary:**

The paper argues that option discovery methods should not attempt to recreate every aspect of a state. Instead, it proposes identifying only the features responsible for novelty and using them to define abstract subgoals. The method combines CFN-based novelty detection with Shapley attribution to select relevant features and construct subgoal classifiers.

**Compliance With Llm Reviewing Policy:**

Affirmed.

**Final Justification:**

The paper presents an interesting and well-motivated critique of state-reaching approaches to option discovery, and the core idea remains original and potentially valuable. However, after considering both the paper and the rebuttal, the reviewer does not find the current submission strong enough for acceptance and therefore maintains the original score. The rebuttal was helpful in clarifying the intended scope of the contribution and addressing several presentation issues. That said, the reviewer’s main concerns remain insufficiently resolved, particularly regarding the strength of the empirical support and the broader applicability of the proposed approach. Overall, while the direction is promising and the clarifications were appreciated, the rebuttal did not materially change the reviewer’s assessment.

**Key Questions For Authors:**

1. This method uses SAM in one experimental setting and a background removal technique in the other. As a result, it appears to operate only in pixel-based environments. It is unclear whether the approach would extend to state-based continuous control tasks with online exploration, such as the state-based Ant Maze. Since those settings do not rely on pixel inputs, it remains a question how the proposed feature-based abstraction mechanism would apply there.

2. The use of external feature extraction tools (e.g., SAM) raises the question of whether meaningful abstractions are truly learned by the RL system or largely rely on pretrained vision models. Because candidate features are provided externally and novelty is defined based on how rarely a state is observed, the discovered subgoals may reflect simple visual differences in segmented objects rather than meaningful or task-relevant structure.

3. The paper uses goal-conditioned R2D2 to train intra-option policies, but provides little explanation of how this learning procedure works or why it was chosen. A brief overview of the training setup and its role in the overall framework would improve clarity and help readers better understand the algorithm.

4. Computational overhead from repeated attribution steps (especially with DeepSHAP) is not discussed, leaving questions about scalability to higher-resolution or real-time settings.

**Limitations:**

yes

**Strengths And Weaknesses:**

**Strengths**

1. The argument that it is unnecessary to revisit every aspect of a state is highly compelling and raises a meaningful and well-motivated problem. The bicycle example in the introduction is also intuitive and effectively conveys the core idea.

2. The paper tackles an important limitation of state-reaching methods and frames the problem in a clear and principled way.

3. The qualitative results, particularly in Montezuma’s Revenge, convincingly illustrate that the discovered subgoals generalize across visually distinct states, supporting the claim of abstraction beyond state-reaching.

**Weaknesses**

1. The overall algorithm can be somewhat difficult to follow. Providing a more consolidated and step-by-step presentation of the full pipeline would improve clarity and make the method easier to understand and reproduce.

2. The method relies on CFN-based novelty, which measures statistical rarity rather than semantic relevance. As a result, visually uncommon but task-irrelevant or uncontrollable changes may trigger subgoal creation. Since Shapley attribution decomposes the CFN signal, any misalignment between novelty and meaningful progress directly propagates into the discovered abstractions.

3. The method introduces several hyperparameters (e.g., in Eqs. 1, 2, and 6), but provides limited justification for their selection and lacks a thorough ablation study. As a result, it is unclear how sensitive the performance is to these choices.

4. As far as I understand, the method appears to be more suited to pixel-based environments than state-based ones, since all experiments rely on either SAM in one setting or a background removal technique in the other.

5. Typo: In Algorithm 1, line 23, the opening parenthesis “(”.

---

> ### Author Rebuttal · Authors · 2026-03-30
>
> We are glad you found our core argument **highly compelling** and **well-motivated**. We also thank you for uncovering gaps in our presentation; these should be fairly easy for us to fix.
>
> > It is unclear whether the approach would extend to state-based continuous control tasks
>
> In fact, our method applies directly to factored state vectors. In that setting, each state dimension is a candidate feature and the rest of the algorithm (attribution, classifier construction, option learning) works as-is. We applied our method to image-based domains precisely because they are the harder case: images require extracting a feature vector from pixels, which is where SAM and contour detection come in. In state-based domains like Ant Maze, that extraction step is already done for us, and the full pipeline from Algorithm 1 applies without modification.
>
> > discovered subgoals may reflect simple visual differences in segmented objects rather than meaningful or task-relevant structure
>
> We realize the paper did not sufficiently clarify an important aspect of our method: subgoals are created from both intrinsic novelty spikes and extrinsic reward spikes. For the extrinsic pathway, we train a binary classifier where positive examples are states that yield higher-than-expected rewards. In sparse-reward domains, this corresponds to positively rewarding events like picking up a key or opening a door. We then apply the same feature attribution algorithm to this classifier as we do for novelty. Figure 4 already shows both pathways — panel (a) decomposes a novelty spike, panel (b) decomposes a reward spike — but we failed to foreground this distinction in the text. Critically, neither pathway alone is sufficient: reward attribution captures task-relevant structure but fires too rarely in sparse-reward domains, while novelty attribution provides the complementary signal needed for exploration. We will make the dual-trigger mechanism explicit in Algorithm 1 and the surrounding exposition.
>
> We also note that the Pixel Equality ablation in Montezuma's Revenge provides direct evidence that SAM alone is not doing the heavy lifting. Pixel Equality uses the same SAM-derived features and the same discovered subgoals as our method, yet performs substantially worse — precisely because it attends to all features rather than the relevant subset. The abstraction comes from our selection mechanism, not the segmentation.
>
> > [novelty] measures statistical rarity rather than semantic relevance
>
> We also respectfully argue that statistical novelty may be a more powerful signal for option discovery than the review suggests. Count-based exploration is provably efficient in the tabular setting (Strehl and Littman, 2008) and has powered some of the most sample-efficient deep RL agents to date (Kapturowski et al., 2022). While novelty can trigger on uncontrollable changes, it is far from a trivial statistical signal. As we discuss in Section 5, pairing novelty with controllability filtering (Bagaria and Schaul, 2023) could address the concern about irrelevant subgoals. Our broader hypothesis is that the combination of novelty, controllability, and reward may serve as a simple and scalable substrate for option discovery; this paper is a small step toward testing that hypothesis.
>
> > Introduces several hyperparameters but provides limited justification and lacks a thorough ablation study
>
> Good point, we are working on these and will try to update you with results soon.
>
> > Not enough discussion about why and how the goal-conditioned R2D2 works
>
> We chose R2D2 with HER because option policies receive sparse binary rewards from the subgoal classifier, and this combination is well-established in goal-conditioned RL. Mechanistically, we modify the R2D2 agent's CNN encoder to include a torso that processes the goal input; the state and goal embeddings are concatenated and fed to an MLP that produces Q-values over actions. For hindsight relabeling, we select 5 subgoals achieved in each episode trajectory and add the relabeled sub-trajectories to the replay buffer. Both choices are standard practice in goal-conditioned RL, which is why we omitted the details from the main text, but we agree that including them would improve clarity and will add this description in the revised paper.
>
> > Computational overhead ... is not discussed
>
> Feature attribution is triggered at most once per episode, and only when the most novel state in the trajectory exceeds the $\mu + \sigma$ threshold, which by definition, happens rarely. When it does trigger, SAM and DeepSHAP run a single forward pass on one image. In practice, the vast majority of episodes do not trigger attribution at all. The overall effect on wall-clock time is marginal compared to the standard computational costs of deep RL—network updates, experience replay, environment stepping—which dominate training time. Thank you for pointing out this omission, we will include this discussion for the camera-ready.

---

> > ### Author Rebuttal · Reviewer_XqSX · 2026-04-03
> >
> > Thank you for the detailed response.
> >
> > However, I believe that the core concerns remain insufficiently addressed.
> >
> > A central weakness of the paper is its reliance on external feature extraction modules such as background subtraction and SAM. While the rebuttal argues, via the Pixel Equality ablation, that segmentation alone is not sufficient, this comparison only demonstrates the benefit of feature selection given the same externally provided feature proposals. **It does not resolve the broader concern that the method’s empirical success may still depend heavily on task-aligned external decomposition.** As a result, it remains unclear how much of the observed performance gain is attributable to the proposed abstraction mechanism itself, as opposed to the privileged feature structure induced by these external modules.
> >
> > More broadly, the current evaluation appears to be specific to environments where novelty can be localized to a small number of object-centric visual features. It remains unclear whether the same pipeline would extend naturally to more dynamic (or egocentric, e.g., Minecraft) visual settings or continuous-control domains, where observations evolve globally and many coordinates jointly contribute to behavior (e.g., where global position in locomotion is governed by the coordinated effect of many low-level joint movements). **In such settings, applying SAM-like modules may be less natural, and the assumption that novelty can be attributed to a small subset of localized features may not hold.** The rebuttal states that the method directly applies to factored state vectors; however, this shifts the burden to the availability of such factorization, which is itself nontrivial in many realistic domains.
> >
> > I understand that there may not have been sufficient time to conduct additional experiments. **Please feel free to provide any further clarification or discussion on these points.**

---

> > > ### Author Response · Authors · 2026-04-06
> > >
> > > We thank the reviewer for their continued engagement.
> > >
> > > - Hyperparameter ablations: We ran ablations on Montezuma's Revenge over two key hyperparameters: σ_state_filtering (novelty threshold for subgoal creation) and σ_patch_filtering (dissimilarity threshold for the patch-based classifier), sweeping each over {1, 2, 3} standard deviations above the mean. Most configurations reach meaningful scores (1000+), confirming the method is not overly brittle to these choices. Lower σ_patch_filtering values consistently perform better---intuitively, a more permissive patch threshold yields classifiers that generalize more broadly and ease option policy learning. For σ_state_filtering, results are more inconclusive, with different values leading at different points during training, suggesting robustness to this choice. Some configurations exhibit instability; this could be because we only had 3 random seeds per configuration due to compute and time constraints, and we will add more seeds for the camera-ready.
> > >
> > > - We want to clarify what our paper claims to contribute and what it does not. Our core contribution is feature selection and its overlooked role in option discovery: determining which features of a novel state are responsible for its novelty. This problem exists even given perfect factored representations, and no prior option discovery method addresses it. We do not claim to solve the representation learning problem (i.e., how to extract factors from raw observations), which remains a major open challenge in RL (Bengio, 2013; Rodriguez-Sanchez et al., 2025; Kumar et al., 2025). We use off-the-shelf solutions (SAM, contour detection) for that step and are transparent about this.
> > >
> > > - Regarding the Pixel Equality ablation: the reviewer states it only demonstrates the benefit of selection "given the same externally provided feature proposals." We agree, and that is precisely the point. Given any set of candidate features, one must still determine which are relevant. That selection step is what enables abstract, transferable subgoals, and it is our contribution. The question of where the candidate features come from is important but orthogonal.
> > >
> > > - Regarding factored state vectors: many widely-used RL benchmarks provide these directly (eg, MuJoCo, DMControl, robotic manipulation). Saying our method applies there is not shifting burden, it is identifying a large and important class of problems where the full pipeline works without external tools. The pixel-based setting is strictly harder and requires additional machinery, which is why we chose it to stress-test our approach.
> > >
> > > - We agree that extending to settings like Minecraft, egocentric vision, or high-dimensional locomotion is important future work. However, novelty estimation and representation learning are themselves major open problems; our method builds on off-the-shelf solutions to both, and as those improve, so will our option discovery technique. We believe that solving one well-defined problem---feature selection for abstract subgoal construction---is sufficient contribution for a single paper.

---

### Official Review · Reviewer_68wP · 2026-03-11

**Soundness:** 2
**Presentation:** 3
**Significance:** 2
**Originality:** 2
**Overall Recommendation:** 2
**Confidence:** 5

**Summary:**

This paper argues that most option discovery methods are too tied to state-reaching and learn overly specific, non-transferable skills. The proposed method uses various heuristics like contour detection and foundation models like SAM to identify which features of a novel state are responsible for its novelty. The method then builds a subgoal classifier that attends only to those features, and trains an option to achieve that abstract subgoal. Since the set of features this classifer attends to is greatly reduced realtive to the full state, the classifier theoretically is less prone to distribution shifts, assuming those shifts occure to features outside the classifiers window of observation. Experiments are run on MiniGrid-KeyCorridor, VisualTaxi, and Montezuma's Revenge, with comparisons to R2D2, CFN, and in Montezuma a state-reaching "Pixel Equality" HRL baseline.

**Compliance With Llm Reviewing Policy:**

Affirmed.

**Final Justification:**

Thank you for the rebuttal, but I remain unconvinced. It does not address the fundamental weaknesses I raised. The claimed novelty is still narrower than presented, the method remains brittle because it relies on domain-specific feature extraction choices, and the evidence for learning reusable options is still weak. Most importantly, the response does not resolve the reliance on a two-stage pipeline in Montezuma's Revenge (the only nontrivial domain tested), which is a substantial limitation for a paper whose central contribution is hierarchical exploration and option discovery.

**Key Questions For Authors:**

1. Please add experiments on stronger option-discovery or hierarchical exploration baselines rather then purely flat RL.
2. Can the authors provide direct quantitative evaluation that the attribution step is actually identifying the right features, rather than relying on qualitative examples?
3. How sensitive is performance hyper parameters such as threshold distances in classifier construction, given that each classifier is built from essentially one positive state? This assumption particularly would seem to fail in environments containing procedural generation.
4. The authors should justify why experiments Montezuma were done in two stages rather then fully online.
5. Have the authors considered experiments on procedurally generated environments such as ProcGen, Craftax, Crafter, NetHack, Minihack, or Minecraft?

**Limitations:**

yes

**Strengths And Weaknesses:**

## Strengths

- The core idea that if novelty comes from only part of the observation, the subgoal classifier should attend to that part of the state only makes intuitive sense.
- The qualitative results showing feature attribution and classifier transfer appear to be strong. The learned classifiers by design ignore irrelevant visual context across different scenarios.
- The Montezuma ablation of Pixel Equality isolates isolates the benefit of attending to relevant features rather than the whole image.

## Weaknesses

- The authors overstate the novelty of the papers contributions. The main novel contribution appears to be the subgoal specific observation extraction. The implementation combining novelty-based exploration, external feature extraction, attribution, and non-parametric classifier construction is perhaps novel in its composition, but mostly an engineering effort and not some algortihmic advance (in addition to being somewhat heuristic).
- On the idea of huristic: each implentation seems overfit to the domain its on: background subtraction and contours; Montezuma uses SAM; complex cases switch to DeepSHAP. This makes the method feel brittle and weakens the claim that the agent autonomously discovers abstractions from raw interaction rather then being carefully engineered by the human designer.
- The classifier construction particularly feels very brittle. Each subgoal classifier is non-parametric, built from a single positive state, and thresholded using a hand-designed distance function. This construction seems ripe for failure in procedurally generated worlds such as Craftax, Crafter, or Minecraft.
- The paper relies too heabily on qualitative evidence for whether feature attribution correctly identifies relevant features and whether classifiers generalize. The single quantitative generalization metric in Montezuma is just the number of unique states where a classifier accepts.
- For a paper about option discovery and hierarchical exploration, I would expect stronger comparisons to more directly related hierarchical or skill-discovery methods. As a minimum, I would expect a compairson to Go Explore.
- The Montezuma experiment's two stage pipline reveals a strong limitation of the paper. Using a two-stage pipeline where subgoals are discovered first and the HRL agent is trained afterward limits the performance of the final hierarchical policy to what is discovered during the first stage. Indeed, for deterministic games like Montazumas revenge, one can simply reconstruct a policy by immitating the best performing trajectories seen during training, i.e. Go Explore.
- I've stated this in previous weaknesses but let me be explicit. Beyond the limitations discussed above. Evaluation on deterministic games like on Montezuma experiment's seems fundimentally limiting if the goal is learning transferrable abstractions like options. In such a scenario, you can perform at best like how Go Explore does. For future versions of this work I would recommend the authors explore experiments in procedurally generated environments such as ProcGen, Craftax, Crafter, NetHack, Minihack, Minecraft etc.

---

> ### Author Rebuttal · Authors · 2026-03-30
>
> We thank Reviewer Kqv5 for their detailed feedback and appreciate their suggestions. We respectfully disagree with several other characterizations and address each concern below.
>
> > The main novel contribution appears to be the subgoal specific observation extraction... mostly an engineering effort and not some algorithmic advance
>
> We disagree. The central argument of our paper is that the field of option discovery has defaulted to state-reaching, and that identifying which features matter for each subgoal is the missing algorithmic step. This is a conceptual contribution about how subgoals should be represented, not an engineering one. The specific tools used to implement each step are modular and swappable---the contribution is the framework of attributing novelty to specific features rather than treating states monolithically.
>
> > Each implementation seems overfit to the domain... This makes the method feel brittle and weakens the claim that the agent autonomously discovers abstractions
>
> The algorithm is the same across all domains: detect a novel state, extract candidate features, attribute novelty to a subset, and build a classifier over that subset. What differs is the feature extractor: contour detection in simple domains, SAM in complex ones. These are routine implementation decisions (like choosing the right CNN architecture for your observation space), not evidence of brittleness.
>
> > Each subgoal classifier is non-parametric, built from a single positive state... This construction seems ripe for failure in procedurally generated worlds
>
> Building a classifier from a single positive example is the problem setting, not a limitation of our design. When the agent encounters a particularly salient state during exploration, that is the only positive example available, and it must determine what made that state salient. This is exactly what humans and other animals do when they encounter something interesting for the first time. Our classifier generalizes from one example precisely because it attends only to relevant features; the quantitative results confirm this, with classifiers firing in 353.8 unique states on average. The reviewer suggests this would fail in procedurally generated environments but does not provide a specific reason why.
>
> > The paper relies too heavily on qualitative evidence
>
> The paper provides quantitative learning curves across all three domains, an ablation isolating the effect of feature selection (Pixel Equality), and classifier generalization statistics in Montezuma's Revenge. The qualitative figures in Figures 3, 4, and 5 supplement these results by illustrating what the method learns, they do not replace the quantitative evaluation.
>
> > Can the authors provide direct quantitative evaluation that the attribution step is actually identifying the right features?
>
> The Pixel Equality ablation provides strong quantitative evidence: it uses the same candidate features and same subgoals as our method, differing only in whether feature selection is applied. The substantial performance gap demonstrates that selecting features matters and that our method selects useful ones. Ultimately, the right quantitative measure of whether the correct features were identified is whether doing so improves downstream task performance, which our learning curves confirm across all three domains. If the reviewer has a specific quantitative metric in mind, we would be happy to consider it.
>
> > I would expect a comparison to Go Explore / one can simply reconstruct a policy by imitating the best performing trajectories
>
> Go-Explore relies on manually defined cell representations and reconstructs policies by imitating best trajectories. It does not discover options or build reusable temporal abstractions. Our paper addresses option discovery, a fundamentally different objective. From our perspective, Go-Explore is another instance of the state-reaching paradigm our work aims to move beyond.
>
> > I would recommend the authors explore experiments in procedurally generated environments
>
> We appreciate the suggestion for future work. However, we note that many valuable and impactful RL algorithms have been developed and published without evaluation in procedurally generated settings; this is not a standard requirement for work on option discovery. Our attribution step operates downstream of the novelty detector: if the novelty estimator generalizes across procedural variations, so will our attribution; if it does not, that is a limitation of the novelty estimator, not of our contribution. We also note that many real-world problems do not involve procedural generation, and additionally, we see no fundamental reason why our approach would not extend to such settings.
>
> > Hyperparameter sensitivity
>
> We acknowledge this gap and are working on ablations over the key hyperparameters

---

> > ### Author Rebuttal · Reviewer_68wP · 2026-04-03
> >
> > As the authors have not followed up to my response I am maintaining my score.

---

> > > ### Author Response · Authors · 2026-04-04
> > >
> > > We did submit a response to your review above, time stamped 31 Mar 2026 at 19:04. Please let us know in case it’s an open review bug and you can’t see the response or if you meant that we didn’t answer a specific question/concern of yours.

---

### Official Review · Reviewer_CEBK · 2026-03-12

**Soundness:** 4
**Presentation:** 3
**Significance:** 3
**Originality:** 3
**Overall Recommendation:** 5
**Confidence:** 3

**Summary:**

Primarily, the work focuses on discovering options an agent can take (abstract sub-goals). To achieve this, the authors combine intrinsic curiosity-based exploration with feature-based novelty estimation using shapely values. Shapely values are computed using DeepSHAP for individual features to discover the sub-goals.

**Compliance With Llm Reviewing Policy:**

Affirmed.

**Final Justification:**

The rebuttal explains why particular experiments are not required and I agree with the argument. The presented work is very clear about its limitations and the rebuttal explains why overcoming the limitations is a hard problem. As such, the work in its limited scope is useful for further research in the sub-field of curiosity driven RL.

**Key Questions For Authors:**

Can the authors provide ablations for Counterfactual reset vs DeepSHAP for marginal novelty calculation?
Can the authors explain why it is not being compared to RND? Or else add it as a baseline?

**Limitations:**

Yes

**Strengths And Weaknesses:**

**Strengths**:

1. The paper is well written, well motivated and situated in theory, and easy to follow
2. The work creates counterfactual features by randomly swapping features from the most novel state and least novel state which allows calculating the marginal novelty instead of DeepSHAP.
3. The features based approach gives some benefits in terms of sample efficiency.

**Weaknesses**:

1. The work needs to rely on external feature extraction methods like contour detection and SAM. The authors do mention this but it very much limits the scope of the work for general environments.
2. There is no ablation on using Counterfactual reset instead of DeepSHAP
3. No comparison with Random Network Distillation (RND) Burda et. al. (2017), which is a well known state reaching baseline.

---

> ### Author Rebuttal · Authors · 2026-03-30
>
> We thank Reviewer CEBK for their thoughtful review and supportive assessment. We address the remaining concerns below.
>
> > Can the authors provide ablations for Counterfactual reset vs DeepSHAP for marginal novelty calculation?
>
> Counterfactual reset swaps individual feature patches between states, which is effective when features contribute to novelty independently, as in MiniGrid and VisualTaxi; this type of observation is quite common in RL domains. In Montezuma's Revenge, however, screen layouts change dramatically between rooms, making patch-level swaps unreliable; DeepSHAP handles this by backpropagating through the novelty network rather than relying on pixel-level correspondence. The two methods suit different visual complexity regimes, and the choice between them is straightforward from domain characteristics alone and no expensive ablation is needed.
>
> > explain why it is not being compared to RND?
>
> The CFN paper (Lobel et al., 2023) showed that CFN performs comparably to RND across most domains including Montezuma's Revenge, with slight improvements in deterministic settings and significant improvements in stochastic ones. CFN is also theoretically more principled, simpler to implement, which is why we chose it as our novelty estimator. More importantly, since CFN and RND perform similarly and our method outperforms CFN by a comfortable margin, we do not expect the relative rankings to change—and the contribution of our paper is the abstract subgoal mechanism, not the choice of novelty estimator.
>
> > The work needs to rely on external feature extraction methods like contour detection and SAM
>
> We agree that feature extraction from pixels is a challenge, but we note that this is a hard open problem in RL more broadly---learning disentangled, factored representations from high-dimensional observations remains an active area of research (e.g., Rodriguez-Sanchez et al., 2025; Kumar et al., 2025). Our method is modular: any method that produces candidate features can be plugged in, and as this line of research matures, our pipeline directly benefits. Crucially, our main contribution is not the feature extraction step but what comes after it---even given a set of candidate features, one still needs to determine which features are responsible for novelty in order to construct abstract, transferable subgoals. The Pixel Equality ablation demonstrates this: same SAM-derived features, same subgoals, but attending to all features rather than the selected subset produces substantially worse performance. Finally, this concern only arises for pixel-based domains---in state-based settings, the factored state vector is already provided and our method applies without any feature extraction at all.
>
> References
> ---------------
>
> Kumar, A., Clune, J., Lehman, J., and Stanley, K. O. (2025). Questioning Representational Optimism in Deep Learning: The Fractured Entangled Representation Hypothesis. ArXiv, abs/2505.11581.
>
> Rodriguez-Sanchez, R., Allen, C. S., and Konidaris, G. D. (2025). From Pixels to Factors: Learning Independently Controllable State Variables for Reinforcement Learning. ArXiv, abs/2510.02484.
>
> Lobel, S., Bagaria, A., and Konidaris, G. (2023). Flipping Coins to Estimate Pseudocounts for Exploration in Reinforcement Learning. In Proceedings of the 40th International Conference on Machine Learning, volume 202, pages 22594–22613. PMLR.

---

> > ### Author Rebuttal · Reviewer_CEBK · 2026-04-03
> >
> > The authors have answered all my questions. I am updating my score to 5.

---

### Official Review · Reviewer_RUWm · 2026-03-12

**Soundness:** 3
**Presentation:** 4
**Significance:** 3
**Originality:** 4
**Overall Recommendation:** 5
**Confidence:** 3

**Summary:**

The paper introduces an option discovery method that addresses the limitations of state-reaching objectives. Existing hierarchical RL methods typically attempt to recreate previously visited target states in their entirety, which limits skill transferability and scales poorly in complex environments. The authors propose an algorithm that leverages novelty-driven intrinsic motivation to find interesting states and then applies feature attribution techniques to isolate the specific subset of features responsible for the state's novelty. The agent then constructs a non-parametric subgoal classifier that attends only to these relevant features, ignoring the rest. This allows the discovered options to generalise across visually distinct states.

**Compliance With Llm Reviewing Policy:**

Affirmed.

**Final Justification:**

The paper proposes a well-motivated option discovery method that uses Shapley-based feature attribution to isolate the subset of features responsible for state novelty. The core hypothesis is sound, the approach is original, and the writing is excellent. My main concerns were hyperparameter sensitivity, the two-stage training procedure in Montezuma's Revenge, and the reliance on external feature extractors. The rebuttal addressed these constructively. While I still believe that the method's scope of applicability remains relatively constrained, the proposed method may be useful for the wider community. I will retain my possitive assessment of the paper.

**Key Questions For Authors:**

1. Could you provide more intuition or ablation data regarding the sensitivity of the hyper-parameters, particularly the novelty threshold ($\mu + \sigma$) used to detect novel events, and the dissimilarity threshold ($\xi$) used in the subgoal classifier?
2. The two-stage training procedure in Montezuma's Revenge limits the evaluation of the algorithm's sample efficiency. Are there fundamental challenges with running your method jointly online in visually complex domains?
3. The feature extraction step currently relies on tools that impose an object-centric prior (SAM, contour detection). How would the feature attribution step behave if the source of novelty is not nicely bounded by these tools (e.g. global illumination changes, shifting background colours)?

**Limitations:**

yes

**Strengths And Weaknesses:**

#### Strengths
+ The core hypothesis, that state-reaching creates an artificial bottleneck in option discovery, is sound.
+ The use of cooperative game theory to calculate the marginal contribution of features to a novelty metric seems to be a good way to identify relevant subgoals.
+ The paper is very well-written and well-motivated.
+ Scaling hierarchical RL for long-horizon, high-dimensional problems is a critical bottleneck in the field; this work takes a meaningful step toward agents that can autonomously build reusable skill repertoires.

### Weaknesses
- The empirical evaluation in Montezuma's Revenge relies on a two-stage training procedure  to stabilise learning. As the authors note too, this makes the comparison against online baselines somewhat rough and limits the ability to evaluate the method's sample efficiency in visually complex domains.
- The necessity of using external feature extractors (like SAM) currently limits the broader significance and plug-n-play utility of the algorithm for non-visual state spaces.

---

> ### Author Rebuttal · Authors · 2026-03-30
>
> We thank Reviewer 68wP for their generous and thoughtful review, and are glad they find the core hypothesis sound, the use of Shapley values well-motivated, and the paper well-written. We are happy to address the remaining questions.
>
> > Could you provide more intuition or ablation data regarding the sensitivity of the hyperparameters?
>
> We are running ablations over the key hyperparameters (novelty threshold, dissimilarity threshold $\xi$, and initiation set threshold $\delta$) and will include them in the revised paper.
>
> > Are there fundamental challenges with running your method jointly online in visually complex domains?
>
> Good question. The core challenge is that novelty estimates are wildly non-stationary early in training. The novelty function is itself being learned, and its non-stationarity destabilizes option learning when both run simultaneously. In simpler domains (MiniGrid, VisualTaxi), the novelty function stabilizes quickly enough that joint online training works well, as our experiments demonstrate. In complex domains like Montezuma's Revenge, this stabilization takes much longer, making the two-stage approach a pragmatic choice---and a common one in HRL research. We are actively working on online integration and view it as an important direction for future work.
>
> > How would the feature attribution step behave if the source of novelty is not nicely bounded by these tools (e.g. global illumination changes, shifting background colours)?
>
> Our feature extraction step is modular---any method that maps observations to candidate features can be plugged in. SAM and contour detection are convenient choices for our current domains, but learning disentangled representations end-to-end is an active area of research (Rodriguez-Sanchez et al., 2025; Kumar et al., 2025), and such methods could serve as drop-in replacements.
>
> > The necessity of using external feature extractors currently limits the broader significance for non-visual state spaces
>
> Interestingly, the relationship is the opposite: our method is actually easier to apply in non-visual state spaces. When a factored state vector is available, each dimension is a candidate feature, counterfactual reset or DeepShap apply directly, and the entire SAM/contour detection pipeline is unnecessary. We chose image-based domains precisely because they stress-test our method in the harder setting.
>
> References
> --------------
> Bengio, Y. (2013). Deep Learning of Representations: Looking Forward. In Statistical Language and Speech Processing, Lecture Notes in Computer Science, vol 7978. Springer.
>
> Kumar, A., Clune, J., Lehman, J., and Stanley, K. O. (2025). Questioning Representational Optimism in Deep Learning: The Fractured Entangled Representation Hypothesis. ArXiv, abs/2505.11581.
>
> Rodriguez-Sanchez, R., Allen, C. S., and Konidaris, G. D. (2025). From Pixels to Factors: Learning Independently Controllable State Variables for Reinforcement Learning. ArXiv, abs/2510.02484.

---

> > ### Author Rebuttal · Reviewer_RUWm · 2026-04-01
> >
> > I appreciate the authors' response. I am satisfied with most of the answers; however:
> > 1. It would be good to get some initial results on hyperparameter sensitivity before I can make a final judgement.
> > 2. I would be prone to dispute the claim that "our method is actually easier to apply in non-visual state spaces". It seems that there is a leap that is being made in your argument -- particularly, the access to factored state representations. While I agree that your method would work with factored states, the core issue stems from the fact that one may not necessarily have access to them (unless, as you mentioned, representation learning methods are applied and are effective). This is what I was getting at specifically. Please feel free to further share your thoughts on this.
> >
> > Thanks!

---

> > > ### Author Response · Authors · 2026-04-06
> > >
> > > Thank you for the continued engagement and we are glad that you are satisfied.
> > >
> > > - On hyperparameter sensitivity: We have now run ablations on Montezuma's Revenge, sweeping σ_state_filtering and σ_patch_filtering over {1, 2, 3} standard deviations above the mean. Most configurations reach meaningful scores (1000+), confirming the method is not overly brittle to these choices. Lower σ_patch_filtering values consistently perform better, as more permissive thresholds yield classifiers that generalize more broadly. Results for σ_state_filtering are more inconclusive, with different values leading at different points during training, suggesting robustness to this choice. We ran 3 seeds per configuration due to compute and time constraints and will add more for the camera-ready.
> > >
> > > - That is correct. When the factors are given (as in many RL benchmarks like MuJoCo or DMControl), our method applies directly without external tools. When factors are not given, additional representation learning is needed. In this paper, we used SAM and contour detection for that step; in the future, factored representation learning techniques could serve as drop-in replacements as they mature further. But yes, we did not mean to imply that learning factored representations is by any means an easy problem.

---

### Decision · Program_Chairs · 2026-04-30

**Decision:**

Reject

**Comment:**

While the abstract frames the primary contribution as utilizing abstracted subgoals rather than standard state-reaching, the manuscript struggles to conceptually and empirically deliver on this premise. As noted by Reviewers 68wP and XqSX, the motivation and core contributions remain difficult to follow. Rather than introducing a novel framework, the proposed algorithm reads as an amalgamation of existing techniques, lacking both a thorough literature review and the rigorous ablation studies required to isolate the source of improvement.

My recommendation leans toward rejection due to the following critical gaps:

* Lack of Targeted Ablations: To validate the core claim, the authors must isolate the impact of deduplicating subgoals that share the same abstractions. Without an ablation that holds the algorithmic backbone constant and only toggles this specific deduplication module, it is impossible to verify if this operation is genuinely meaningful.

* Questionable Generalizability: The method relies on a highly specific sequence of non-standard operations—such as employing Shapley Value-related techniques to identify representative features via a pre-trained coin-flip network. The manuscript fails to investigate whether the observed improvements represent a generalizable phenomenon or are merely an artifact heavily tethered to this idiosyncratic setup.

Ultimately, because the manuscript lacks the rigorous baselines and systematic empirical validation required to substantiate its central claims, it does not meet the threshold for acceptance in its current form.